# Magnetic effects of thaumatin crystals; observation of crystal growth by magneto-Archimedes levitation and magnetic orientation

Syou Maki[1]*, Masao Fujiwara[2], Yoshihisa Fujiwara[2], Makoto Nakabayashi[3], Shotaro Morimoto[3], Seiichi Tanaka[4], Seiji Fujiwara[4], Masayuki Hagiwara[5], Kohji Inaka[6]

1 Institute of Frontier Science and Technology, Okayama University of Science, Okayama, Okayama Pref, Japan, 2 Program of Mathematical and Life Sciences, Graduate School of Integrated Sciences for Life, Hiroshima University, Higashihiroshima, Hiroshima Pref., Japan, 3 Faculty of Pharmacy, Laboratory of Molecular Chemistry, Osaka Ohtani University, Tondabayashi, Osaka Pref., Japan, 4 Department of Mechanical Engineering, National Institute of Technology, Akashi College, Nishioka, Akashi, Hyogo Pref., Japan, 5 Center for Advanced High Magnetic Field Science, Graduate School of Science, Osaka University 1–1, Toyonaka, Osaka Pref., Japan, 6 Maruwa Foods and Biosciences, Co., Ltd., Yamatokoriyama City, Nara Pref., Japan

* makisyou@ifst.ous.ac.jp

**Data Availability Statement:** All relevant data for this study are publicly available from the Okayama

## Abstract

We examined the magnetic effect of thaumatin crystals, which are a well known model of protein crystals but which have hardly been studied for that effect. We succeeded in crystallizing thaumatin by magnetic levitation based on the magneto-Archimedes effect by the addition of the paramagnetic substance gadolinium chloride. We also carried out a chronological observation of the levitation process in a superconducting magnet, and visualized the magnetic orientation of the crystals by applying a magnetic field along the horizontal direction. In another major result, we carried out a diffraction experiment and performed a structural analysis of the crystals. We noticed from the results that no electron density from the gadolinium ion could be observed in the crystals. This suggests the possibility that the paramagnetic substance of the aqueous precipitant solution helps only to promote the crystals' levitation, and has little effect on thaumatin crystallization.

## Introduction

Thaumatin is an industrially useful protein that is characterized by being 3,000 times sweeter than sucrose [1–5]. Purely extracted thaumatin crystallizes easily and often grows to a size large enough to be confirmed by the naked eye. Due to these favorable characteristics, thaumatin is well known as a model protein frequently used in various academic fields of research, especially in structural biology. Crystallographic details of thaumatin are made public in the Protein Data Bank (see the data of PDB entry 1RQW). Among the many protein crystals, we are deeply interested in the feature of the crystals growing to a large size, because large sized

University of Science repository (https://ous.repo.nii.ac.jp/records/2000296).

**Funding:** MEXT/JSPS, KAKENHI, Grant Number JP15K04669 and Grant Number JP20K04335. Institute of Frontier Science and Technology, Okayama University of Science. Center for Advanced High Magnetic Field Science in Osaka University under the Visiting Researcher's Program of the Institute for Solid State Physics, the University of Tokyo. The funders had no role in study design, data collection and analysis, decision to publish, or preparation of the manuscript.

**Competing interests:** Every author declares no competing interests.

protein crystals are essentially important for adopting in measuring by the transient short hot-wire method [6,7]. We previously used this method to successfully measure for the first time the thermal properties (thermal conductivity and thermal diffusivity) of hen egg white lysozyme (HEWL) crystals [8–10]. We think that thaumatin is a preferable protein worth adopting as our next measurement target. If the thermal properties of thaumatin can be compared with those of HEWL, instructive information on heat transfer of other protein crystals may be distinguished by using a molecular structural perspective. Such instruction will be useful for the development of crystal growth technology because we notice that temperature control has a significant impact on the crystallization [11] and thermal properties can make a contribution to the progress of temperature control. Growing protein crystals by levitation has an advantage other than the measurement of thermal properties. Generally, protein crystals are heavier than the solution, the crystals grow at the bottom of the container and often adhere to the bottom wall. The gas-liquid interface is a containerless condition [12–16], and it is expected that high-quality crystals will be produced. By growing them at the interface, we can easily pick them up without damaging them, which is convenient for structural analysis. Control over thaumatin crystallization by a magnetic force [17] is indispensable to the accomplishment of that measurement. In comparison with HEWL crystals [18], however, only a few magnetic effects of thaumatin on crystallization have been examined yet [19–24]. Previous research has only confirmed the presence of the magnetic orientation of thaumatin crystals, by applying a high magnetic field of 10 or 11 T and growing the thaumatin crystals in a small capillary tube enclosed with gel [19,20]. Its other magnetic effects, such as the magneto-Archimedes effect [25], remain unknown. A possible reason for why the magnetic effect of thaumatin crystals has not been investigated until now may be the fact that a stronger magnetic field would be required to observe an apparent effect than is the case with HEWL crystals. Therefore, we performed experiments with a superconducting magnet that generates a very strong magnetic force. The stronger the magnetic force, the more easily the effect can be observed with a small amount of paramagnetic substance, especially in employing magneto-Archimedes levitation. If magnetic levitation can be realized in thaumatin crystal growth, it will be proven that our method, which has only been successful with HEWL so far, is applicable to the crystallization of other proteins.

We additionally examined the structural analysis of the thaumatin crystal, and investigated the effect of the paramagnetic substance in the precipitant agent through the structure refinement. As far as we know, there has been no previous research on using a paramagnetic substance to crystallize thaumatin, and it is also unknown whether or not such substances can bond to thaumatin molecules. Our investigation will provide detailed insight into the effect of the paramagnetic substance on magnetic levitation. We developed all of the observation equipment used in this study, and here we report this approach and these techniques in detail.

## Theoretical ideas

In this study, we shall name the system in which the bore of a superconducting magnet is vertical as "*a vertical bore system*", and the system in which it is horizontal as "*a horizontal bore system*". The magnetic force vector $\boldsymbol{F}(f_r, f_\theta, f_z)$ is defined by a cylindrical coordinates system. Here $f_r, f_\theta$, and $f_z$ are the radial, circumferential, and axial components, respectively. Note that $f_\theta$ is theoretically zero in a cylindrical force distribution from a solenoidal magnet.

We adopted a vertical bore system in our experiment on magnetic levitation, and utilized a horizontal bore system for the magnetic orientation, because a horizontal bore system is preferable for visibly confirming the crystals' orientation. In the latter system, a horizontal magnetic field was applied to the crystals and we observed the crystal growth from above, referring

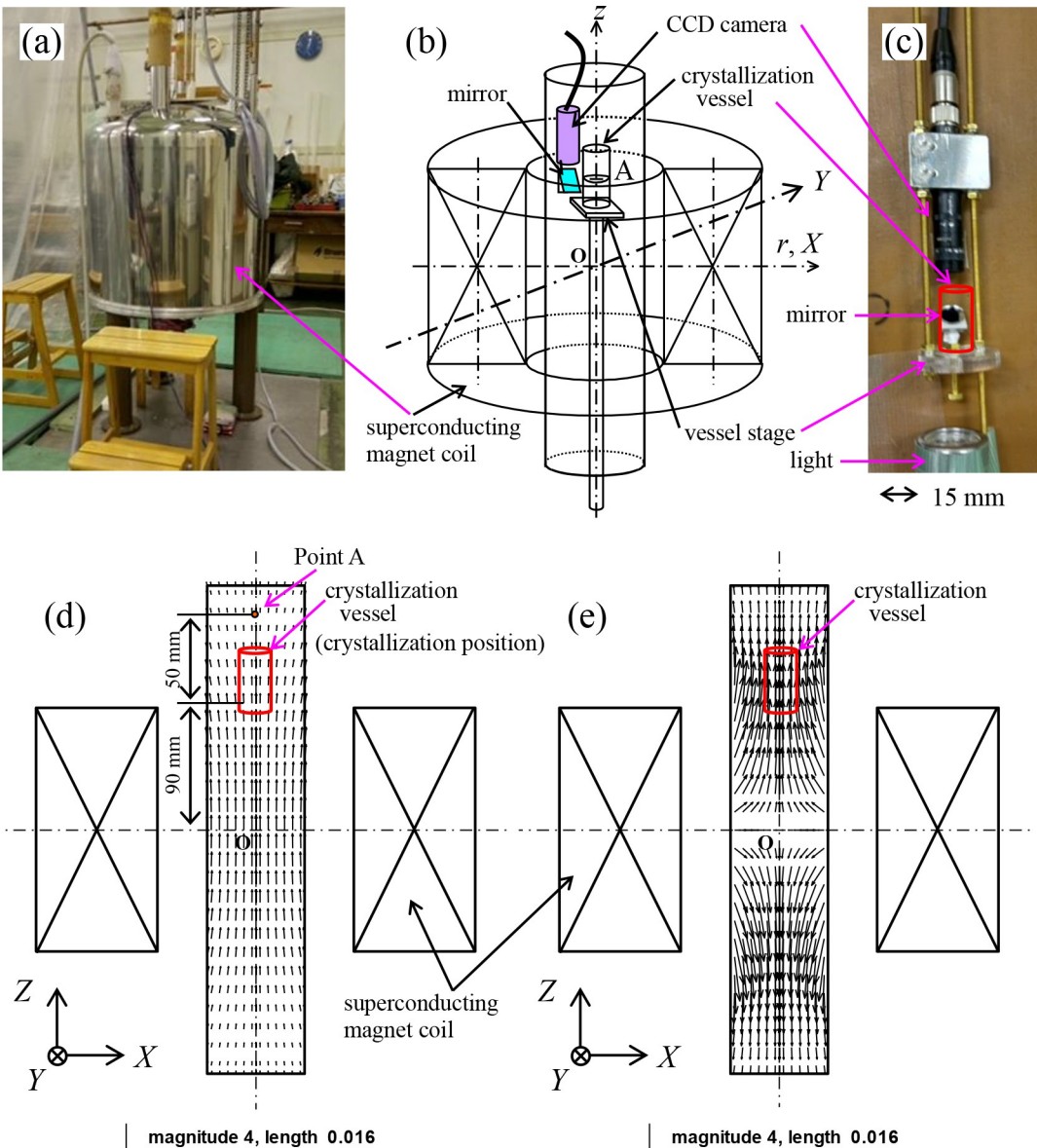

**Fig 1. Superconducting magnet used in the experiment and a schematic illustration of the vertical bore system of the magnet.** (a) A photograph of the magnet, (b) a schematic illustration of the vertical bore system, (c) the support stage for fixing the crystallization vessel, (d) the numerical calculation results of the magnetic field vector **B**, and (e) the results of the numerical calculation of the magnetic force vector **F**, formed inside the bore. Both distributions were obtained by the calculations using an original calculation code developed by the authors. In this program, the magnetic flux density is calculated using the Biot-Savart law. The crystal growth was observed in situ using a CCD camera and a mirror. The crystallization vessel is schematically shown in the red rectangle in (c), which is set up 90 mm above the magnet center O, as shown in (d) and (e). We shall name that location the "crystallization position" in this paper. Point A is located about 50 mm higher than the crystallization point in the vessel.

to the method used in our previous study [26]. The photos and schematic illustrations of these systems are exhibited in Figs 1 and 2. Note that both systems were designed with reference to our previous systems that were used for different purposes. In Fig 1, we define the vertical upper direction as positive.

## Magneto-Archimedes levitation

The magneto-Archimedes levitation of the diamagnetic substance of protein crystal is enhanced by the repulsion of the magnetic forces, which act on the difference between the paramagnetic substance of the solution and the diamagnetic substance of the crystals. We added gadolinium chloride ($GdCl_3$) to the buffer solution and crystallized thaumatin by the batch method. The gadolinium ion ($Gd^{3+}$ ion) possesses large paramagnetic susceptibility and is suitable for conducting magnetic levitation, even in the condition of a weak magnetic field [27]. The success in the levitation of HEWL crystals is attributed to the use of $GdCl_3$ as a precipitant agent [28,29]. As far as we know, no protein crystals other than HEWL crystals have been successfully grown by magneto-Archimedes levitation.

More particularly, the magneto-Archimedes effect is exerted not only to levitate crystals vertically by the component $f_z$, but also to move them horizontally by the component $f_r$. We confirmed that the effect of $f_r$ could become tangible through the techniques of magnetic separation in a weightless condition [30] as well as in an experiment for the thermal properties of HEWL crystals [31]. For successfully realizing a "containerless condition (or contactless condition)" [12–16] in protein crystal growth, it is necessary to make fine adjustments to the balance of forces among $f_r$, $f_\theta$, and $f_z$. The technique of a "magnetic force booster [32,33]" may often be required to realize a "perfect containerless condition" [34,35]. When applying a transient short hot-wire method in a future study, however, it will be enough to conduct simple crystal growth of thaumatin levitated on the gas-liquid interface of the solution, where the probe used in the method is fixed in advance. In this study, we demonstrate a levitating crystallization of thaumatin where the $f_r$ is negligibly small compared to the $f_z$. Owing to this simplification, the driving force of the crystallization in a vertical bore system is presented as:

$$f_z = -(\rho_c - \rho_s)g + \frac{(\rho_c\chi_c - \rho_s\chi_s)}{\mu_0}b_z\frac{\partial b_z}{\partial z} \tag{1}$$

Here, $\rho_c$, $\rho_s$, $g$, $\chi_c$, $\chi_s$, and $\mu_0$, are the crystal density (kg/m$^3$), solution density (kg/m$^3$), gravitational acceleration (m/s$^2$), the mass magnetic susceptibilities (m$^3$/kg) of the protein crystal, the mass magnetic susceptibilities (m$^3$/kg) of the protein solution, and magnetic permeability of vacuum (H/m). $b_z$ is axial components of magnetic flux density vector $\boldsymbol{B}$, and $\boldsymbol{B}$ is defined as $\boldsymbol{B} = (b_r, b_\theta, b_z)$ (T). The first term of Eq 1 represents the effect of buoyancy, and the second term corresponds to the magnitude of the magnetic force. The vertical upper direction was defined as positive.

Magneto-Archimedes levitation of the crystals is explained as follows. Since the solution contains the paramagnetic substance of $Gd^{3+}$ ion, $\chi_s$ changes to paramagnetic (the sign is positive). On the other hand, $\chi_c$ is diamagnetic (the sign is negative), thereby the sign of $\rho_c\chi_c - \rho_s\chi_s$ becomes negative. The sign of $b_z\frac{\partial b_z}{\partial z}$ becomes negative when the crystal-growth position is set higher than the magnet center. Therefore, the second term of Eq 1 becomes positive and the crystal levitation with vertically upward direction is enhanced. In actuality, Eq 1 deduces some practical merits that most of the gravity on the crystal can be canceled simply by the buoyancy force. In the case of HEWL crystallization, about 77% of the crystal weight is cancelled by the buoyancy [10]. Generally, upward forces originating from the buoyancy force are inevitable for actualizing the magnetic levitation of protein crystals.

## Experimental system

### Equipment for the magneto-Archimedes levitation

Fig 1A shows a photograph of the superconducting magnet (JMTA-15T40, Japan Superconductor Technology Inc. (JASTEC)) used in the experiment, and Fig 1B is a schematic

illustration of the vertical bore system of the magnet. Fig 1C presents a support stage for fixing the crystallization vessel. The crystal growth was observed in situ using a CCD camera (CS9300, Toshiba Teli Corporation) and a mirror. All the crystallization images were stored by a video recorder (GV-D1000, Sony Co. Ltd.).

Fig 1D and 1E show the numerical calculation results of the magnetic field vector $B$ and the magnetic force vector $F$, formed inside the bore. The distribution of the magnetic flux density vector $B$ and the magnetic force vector $F$ were obtained by the calculations using an original calculation code developed by the authors. In this program, the magnetic flux density is calculated using the Biot-Savart law [36]. Details are explained in previous studies [37,38]. The direction of $F$ corresponds to the cases acting on the diamagnetic substance. The crystallization vessel schematically shown in the red rectangle in Fig 1C was set up at a location 90 mm above the magnet center O (see the red rectangles in Fig 1D and 1E). We shall name the location the "crystallization position" in this paper. When applying a magnetic field, the paramagnetized buffer solution placed at the crystallization position is forced vertically downward by the magnetic force, and its repulsive force simultaneously works on the crystals to levitate them vertically upward in the solution. This is denoted in the second term of Eq 1. Note that the magnetic levitation of the crystals is possible even when the crystal is non-magnetically oriented.

## Equipment for the magnetic orientation

In most previous studies, magnetic orientation of thaumatin crystals was examined in a small capillary tube [19,20]. In this study, we crystallized thaumatin by the so-called "hanging drop method", in which a droplet is adhered to the bottom of a cover glass. Owing to that method, thaumatin crystals could grow in a droplet keeping a containerless condition. The magnetic field was applied to the droplet along the horizontal direction, and the crystal growth was observed from the top view of the droplet through the cover glass. Fig 2A shows a photograph of the inclined superconducting magnet (JMTD-6T100EF3, Japan Superconductor Technology Inc. (JASTEC)). Fig 2B represents the visualization system installed in the magnet, with the combination of (a) a crystallization vessel, (d) a CCD camera (022MINI-5M-f12-DS, Artray Co., Ltd.), (e) a mirror, (f) a T-type thermocouple (Anbe SMT Co., Ltd.), and (g) a light (LA-100USW, Hayashi-Repic Co. Ltd.). Fig 2C exhibits a schematic illustration of the horizontal bore system. Fig 2D presents the computational distribution of the magnetic field in the bore. The vessel ((a) in Fig 2B and 2C) was set up in the vicinity of the magnet coil center, where the strongest magnetic flux density was 6.0 T. The bore diameter was 100 mm, but the actually available space was a diameter of 70 mm or less in a narrow circular space of the magnet bore because the temperature was controlled by inserting a hose ((b) in Fig 2A and 2C), in which constant-temperature water circulated from a thermostatic bath ((c) in Fig 2B). For that reason, a CCD camera ((d) in Fig 2B and 2C) was fixed horizontally and a mirror ((e) in Fig 2B and 2C) was modulated to observe the crystal growth from above.

## Crystallization

We optimized the crystallization condition before the experiment. Crystallization of thaumatin was performed by a direct mixing method. All we had to do was to mix three types of solutions, (a) protein, (b) buffer, and (c) precipitant, in the optimal volume ratio. Those screening data are summarized in Tables A and B in the Appendix in S1 File. An aliquot of 175.0 μL of (a) thaumatin solution with a concentration of 25 mg/mL (2.5%(w/v)) was mixed with 283.5 μL of (b) the solution containing 0.2 M $N$-(2-acetamido) iminodiacetic acid-NaOH buffer (pH 6.5)

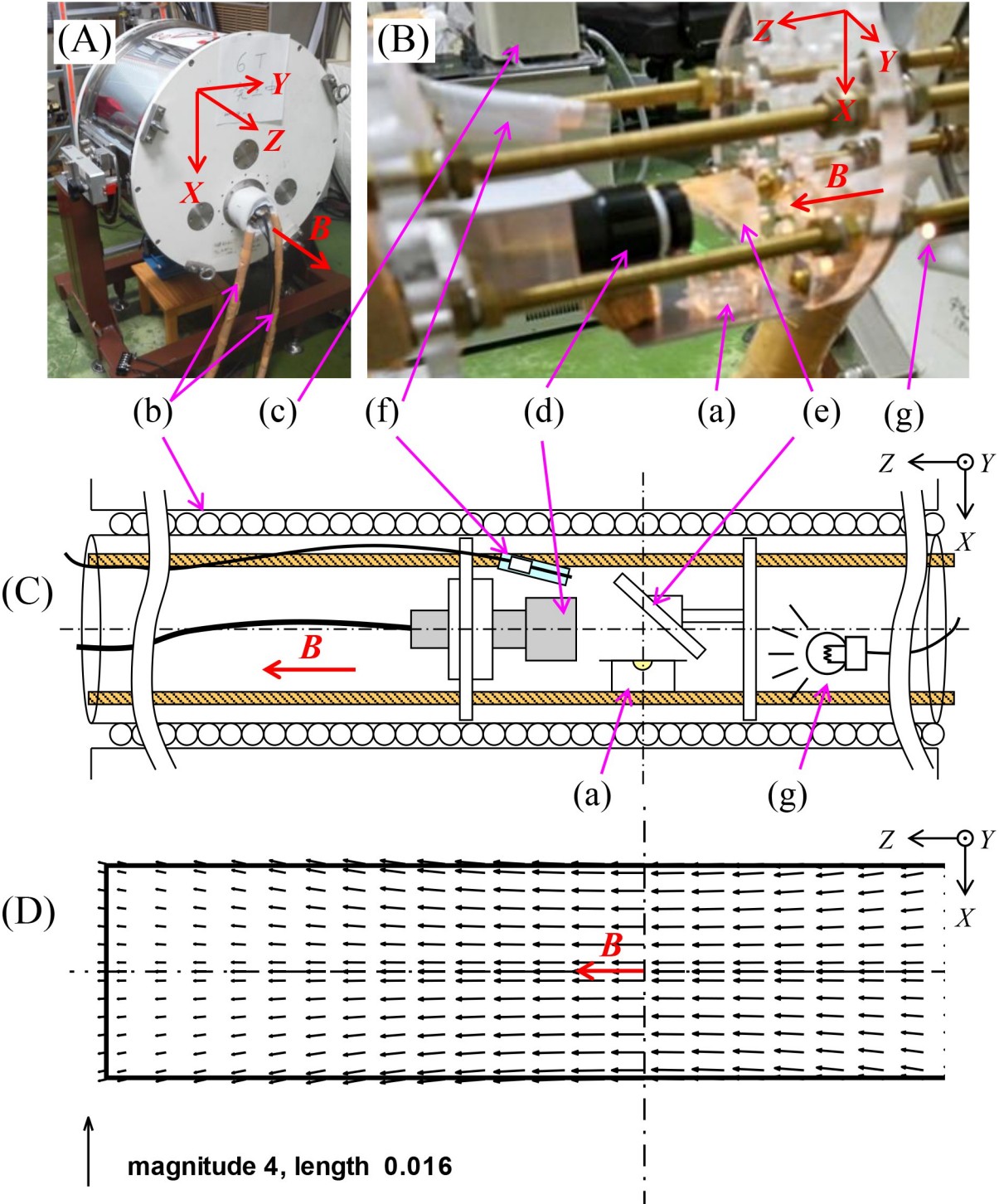

**Fig 2. The horizontal bore system: (A) shows a photograph of the inclined superconducting magnet used in our experiment.** (B) represents the visualization system installed in the magnet. (C) exhibits a schematic illustration of the system. (D) presents the computational distribution of the magnetic field vector ***B***. This was obtained by using an original calculation code developed by the authors as well as Fig 1. (a) A crystallization vessel set up in the vicinity of the magnet coil center where the magnetic flux density was the strongest, (b) a hose through which water of a constant temperature circulates, (c) a thermostatic bath to control the temperature, (d) a CCD camera, (e) a mirror, (f) a T-type thermocouple, (g) a light. The thaumatin droplet was hung from the ceiling of a cover glass, similar to the hanging drop method, and sealed by putting it on the top of the crystallization vessel. The photograph was taken automatically from the top through the transparent cover glass.

and 1.2 M potassium sodium tartrate and 0.04% $NaN_3$. Next we added 70.0 μL of (c) the aqueous precipitant solution of 0.4 M $GdCl_3$ to the aforesaid buffer solution.

The crystallization was conducted with and without a magnetic field of 6.0 T. The magnetic conditions of $b_r$, $b_\theta$, $b_z$ and $b_z \frac{\partial b_z}{\partial z}$ in the bore were referenced in the performance table attached to the magnet. The temperature condition was kept at 20˚C. It took three days to obtain crystals of X-ray diffraction quality.

We used crystals precipitated in the absence of a magnetic field to confirm in advance whether the crystals would magnetically levitate, then we observed the magnetic levitation of those crystals with the CCD camera in Fig 1C. The levitation is highly reproducible regardless of the deterioration of protein, because it depends simply on the magnetic susceptibility and the magnetic field strength. Owing to such preliminary experiments, we were able to achieve highly reproducible levitating crystal growth.

## Results

### Magnetic levitation of thaumatin crystals

The crystallization was carried out through the vertical bore system. The magnetic field gradient $b_z \frac{\partial b_z}{\partial z}$ was 1100 T²/m in the vicinity of the upper coil edge. The magnetic flux density corresponded to about 14.1 T at the magnet center. The temperature in the bore was kept at 20˚C. Fig 3 shows the chronological change in the crystal growth as observed from the vessel side. Fig 3A is a photograph immediately after the vessel was positioned into the bore. The curved meniscus of the non-horizontal gas-liquid interface indicates that the magnetic force driving

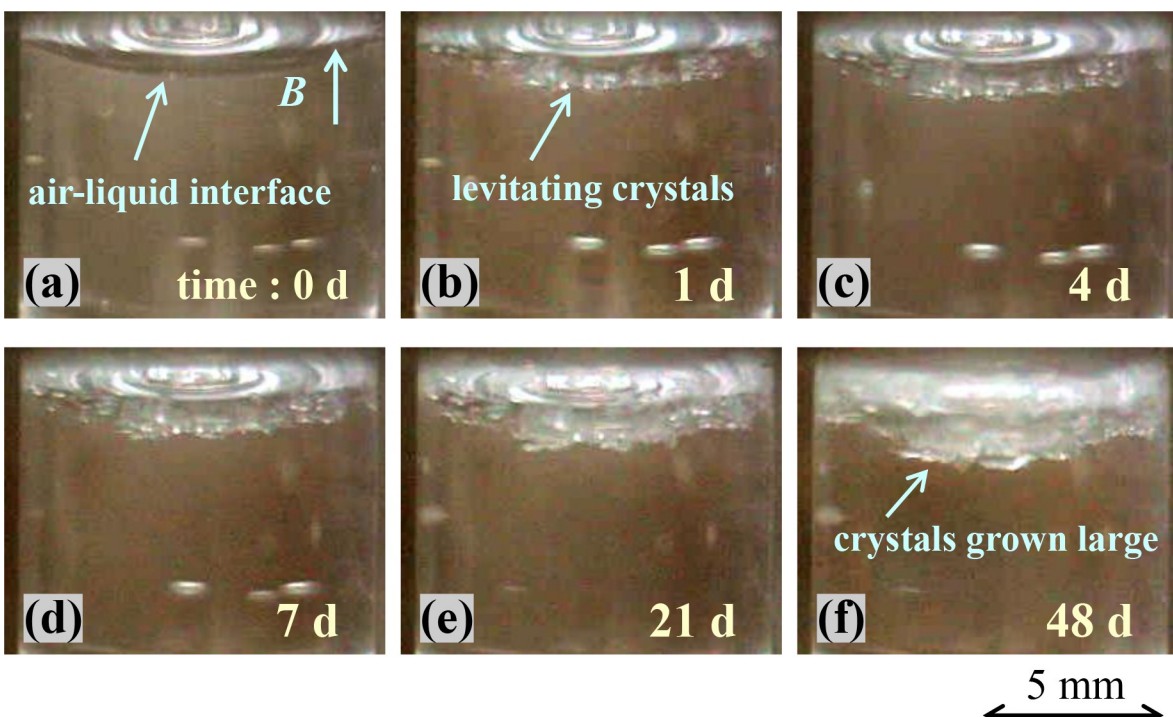

**Fig 3. Chronological change of the crystal growth of thaumatin in the vertical bore system.** The crystal growth was carried out at 20˚C and was observed from the vessel side. (a) A photograph immediately after positioning the vessel into the bore, the crystals at (b) one day later, (c) four days later, (d) seven days later, (e) 21 days later, and (f) 48 days later. From these photos, we have proven that thaumatin crystals were grown as levitating at the gas-liquid interface by the magneto-Archimedes effect. The maximum thickness of the thaumatin crystals exceeded 1 mm after 48 days.

the paramagnetic solution downward was not very strong. In the case of lysozyme levitation, the concentration of paramagnetic substance in the solution was high, so the solution was strongly pulled downward by the magnetic force, causing the meniscus to deform and became almost horizontal in the magnetic field [28]. Fig 3B shows the growing thaumatin crystals at the meniscus one day later. They were approximately 0.1–0.2 mm in size by appearance. This image presents clear evidence that it is possible to levitate thaumatin crystals by the magneto-Archimedes effect, similar to the mechanism in HEWL crystals. Fig 3C shows the state after four days, revealing that many crystals appeared at the interface as growing. They were approximately 0.3–0.5 mm in size. Fig 3D–3F are the images after 7, 21 and 48 days, respectively. It was difficult to measure the size of each crystal. The thickness of the crystals further increased to be as thick as to cover up the whole meniscus. Based on these images, the thickness in Fig 3D and 3E were about 0.8 and 0.9 mm, respectively, and the maximum thickness in Fig 3F exceeded 1 mm after 48 days. The crystal growth rate was a maximum of 0.2 mm/d up to the first day, and a maximum of 0.1 mm/d from the first to fourth days, but then gradually slowed down, reaching a maximum of about 0.01 mm/d from the 21st to 48th days. On the other hand, almost no crystals adhered to the vessel sidewall. As to the magnetic orientation on the crystals, we could not find any evidence through observation from the viewpoint side.

We advanced a video-recording of that levitation process of thaumatin crystals by changing the magnitude of the magnetic force. Its magnitude was changed by moving the vessel up and down about 50 mm from the crystallization position. We shall name the top point as point A (see Fig 1D). The series of photographs in Fig 4 exhibit the crystals' levitation recorded in real time speed. When the vessel was elevated to point A, the crystals sank to the vessel bottom, as can be seen in Fig 4A because the magnetic force was weakened at A. As the vessel was returned to the crystallization position and the magnetic-force magnitude was recovered, the crystals levitated to the meniscus. We could observe the moment when the crystals levitated, as shown in Fig 4B and 4F. These observations were performed at 20°C. This short movie (35 seconds long and named as "Short movie of Fig 4") also reveals the evidence that we could magnetically control the levitation of the thaumatin crystals. We will make this movie available as supplemental data of this research.

## Magnetic orientation of thaumatin crystals

As aforementioned, the magnetic orientation was observed through the horizontal bore system. The solution was the same as that used for conducting the magnetic levitation. A droplet of this solution (about 3 µL) was set beneath the ceiling of a cover glass, similar to the hanging drop method. The cover glass hanging the droplet was then placed on the top of the crystallization vessel (1000 µL), as shown in Fig 2B and 2C. We installed the solution in the vessel bottom by mixing it with 20 µL of 0.4M GdCl$_3$ solution and 81 µL of the precipitant solution as a reservoir. Finally, we sealed the vessel to prevent it from drying.

The photographs of the crystal growth were taken automatically from the top through the transparent cover glass, as schematically shown in Fig 2C. All the images were automatically stored to a PC every 10 minutes. The vessel was positioned in the vicinity of the magnet center, and the magnetic flux density of 6.0 T was horizontally applied as shown in Fig 2D. The temperature in the bore was maintained at 20°C and automatically measured using the T-type thermocouple ((f) in Fig 2B and 2C). The temperature data was taken by using a data logger (NR-1000, Keyence Co., Ltd.). The equipment in the bore was illuminated by the light ((g) in Fig 2B and 2C) from the angle perpendicular to the viewing direction.

Fig 5 shows the chronological change of the crystal growth observed from the vessel top in the horizontal bore system. In Fig 5, the magnetic flux is directed upward in the photo. Fig 5A

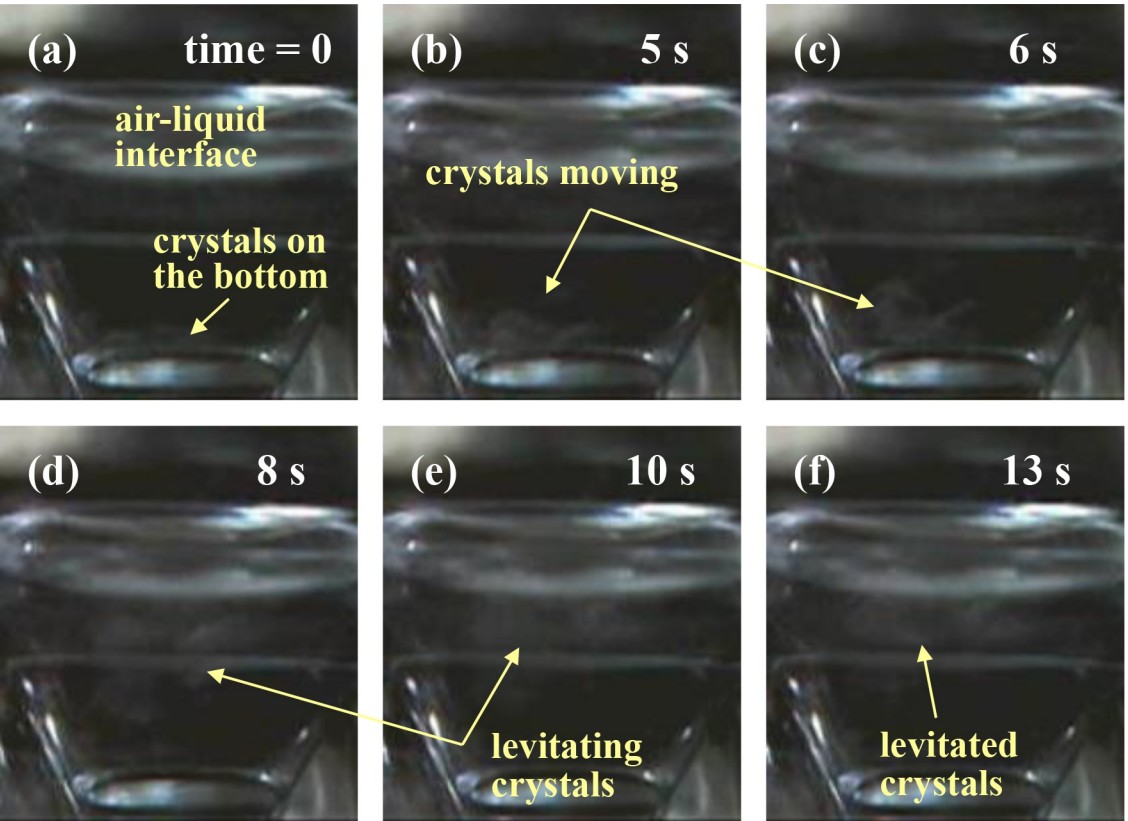

**Fig 4. The magnetic levitation of thaumatin crystals by changing the magnitude of the magnetic force.** The observation was performed at 20˚C. (a) Initial state of 50 mm above the crystallization position. The crystals sank to the vessel bottom. Upon returning the vessel to the crystallization position, the levitating force on the crystals was recovered and we could observe the moment when thaumatin crystals levitated. These observations were performed at 20˚C. (b) 5 s, (c) 6 s, (d) 8 s, (e) 10 s, and (f) 13 s from the initial state. We will make available a supplemental short movie of this process named "Short movie of Fig 4", which was recorded at actual speed.

is a photograph taken immediately after inserting the vessel into the bore. Fig 5B represents the crystallization after four hours. We can see that small crystals have appeared. Fig 5C shows the stage after six hours, and rugged crystals have grown larger and more distinct. These crystals look like a triangle, but are actually an octahedron or a bipyramid. Fig 5D is the state after 20 hours. The size and number of crystals had changed little from the state they were in after six hours. Fig 5E is an enlarged photograph of the white rectangle in Fig 5D. These photographs provide evidence that many bipyramidal crystals were oriented with respect to the direction of the magnetic field $b_z$. Since all the crystals were grown oriented from the beginning, the magnetic orientation of thaumatin is found to be possible at 6 T, smaller than 10–11 T. Fig 5F is a schematic illustration of the crystals' orientation. We assume that the levitated crystals must be oriented, although the presence of the orientation cannot be observed in the photograph of Fig 3.

The supplemental short movie presenting the whole crystallization process of the magnetic orientation edited at 20,000 times speed is available in this journal. See the file of "Short movie of Fig 5".

In summary, we found the following magnetic effects of thaumatin crystal through the experiments: (1) we succeeded for the first time in the magneto-Archimedes levitation of

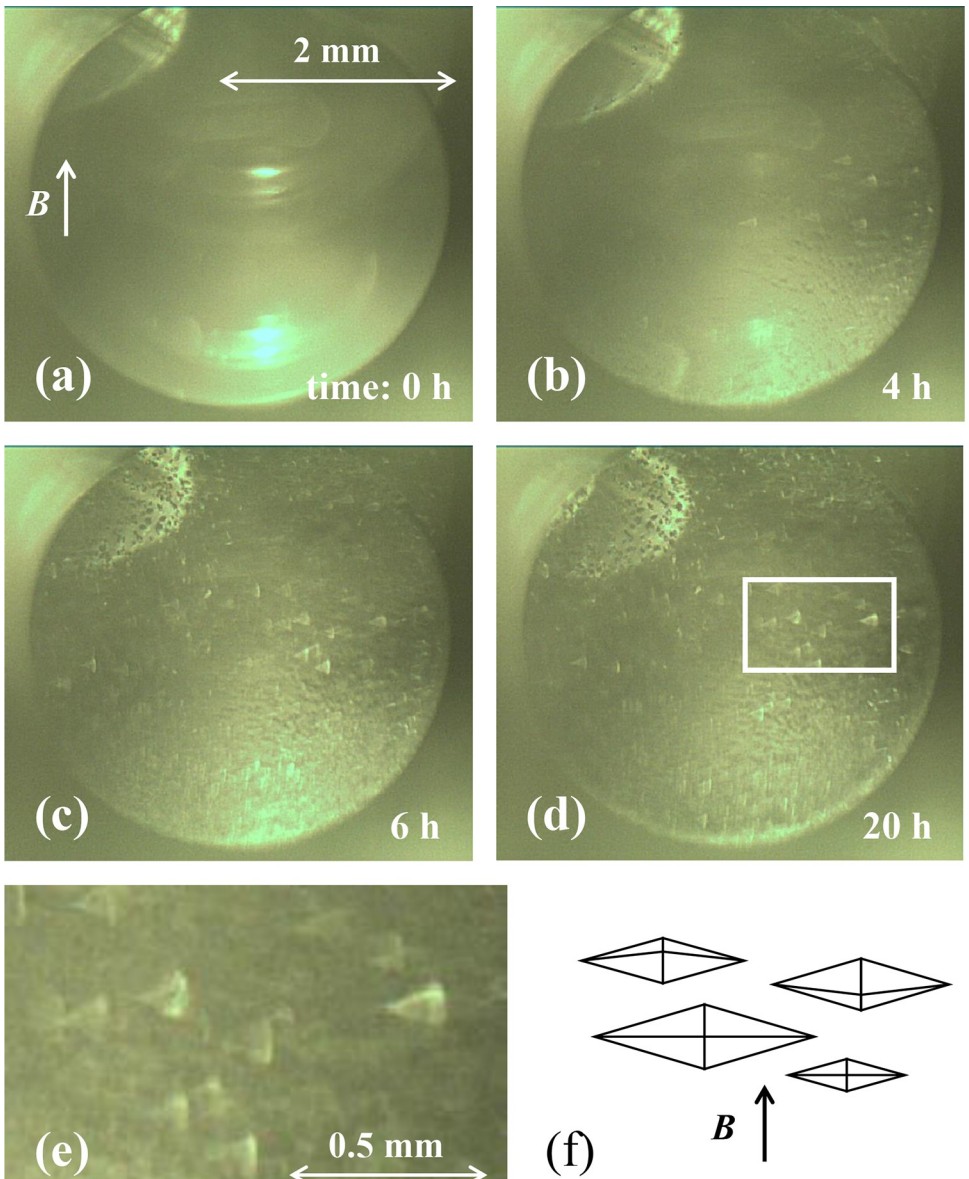

**Fig 5. The chronological change of the crystal growth observed from the vessel top in the horizontal bore system.**
The temperature in the bore was maintained at 20°C. (a) A photograph immediately after positioning the vessel into the bore, the crystals at (b) four hours later, (c) six hours later, and (d) 20 hours later. (e) An enlarged photograph of the white square in (d). It is evident that the crystals are oriented with respect to the direction of the magnetic field. (f) A schematic illustration of the crystals' orientation. We can see that many bipyramidal crystals were oriented with respect to the direction of the magnetic field $b_z$. We will make available a supplemental short movie of this process named "Short movie of Fig 5", which was edited at 20,000 times speed.

thaumatin crystallization; (2) we visualized the crystallization process of the orientated thaumatin crystals in a magnetic field.

## Structural analysis

The thaumatin crystals were examined by structural analysis. To begin with, the selected crystals, grown in the magnetic field, were picked out and immersed in a cryoprotectant solution

**Table 1. Crystal quality by X-ray diffraction.**

| | |
|---|---|
| Diffraction source | SPring-8 BL45XU |
| Wavelength (Å) | 0.7750 |
| Detector | PILATUS 6M |
| Total rotation range (˚) | 480 |
| Oscillation range (˚) | 0.1 |
| Space group | $P4_12_12$ |
| $a$, $b$, $c$ (Å) | 58.08, 58.08, 150.47 |
| α, β, γ (˚) | 90.00, 90.00, 90.00 |
| Resolution range (Å) | 45.98–0.99 (1.00–0.99) |
| Total no. of reflections | 1,196,640 (56,831) |
| No. of unique reflections | 145,471 (7,116) |
| Completeness (%) | 99.0 (100.0) |
| Multiplicity | 8.2 (8.0) |
| Mn($I$) half-set correlation ($CC_{1/2}$) | 0.998 (0.930) |
| $\langle I/\sigma(I)\rangle$ | 20.6 (3.5) |
| $R_{r.i.m.}$ | 0.055 (0.652) |
| $R_{p.i.m.}$ | 0.026 (0.314) |
| Overall $B$ factor from the Wilson plot (Å$^2$) | 8.176 |

Values in parentheses are for the outer shell.

containing 60 mM GdCl$_3$ and 17% glycerol in the mother liquor. The soaked crystals were collected using a Cryo-loop (Hampton Research, Aliso Viejo, CA, USA) and were frozen quickly by immersing them into liquid nitrogen. These frozen crystals were applied to the data collection that was conducted in a synchrotron radiation institute (SPring-8, Japan). In actuality, X-ray diffraction data for single-crystal measurements were collected at 100 K in a stream of nitrogen gas, using a PILATUS 6M detector (Dectris, Baden-Daettwil, Switzerland) at the BL41XU and BL45XU beamlines. All the processes of crystal-centering, data-collection, and processing were conducted automatically by the program suite ZOO [39,40]. In this study, the crystals were helically rotated by 480˚ with an oscillation angle of 0.1˚ per frame.

The structure was solved by molecular replacement with the program MOLREP [41], and finalized sets of atomic coordinates were obtained after iterative rounds of model modification with the program COOT [42], refined with the programs REFMAC5 [43] and PHENIX [44] by rigid body refinement, simulated annealing, positional minimization, water molecule identification, occupancy and anisotropic B-value refinement. The Ramachandran plot of the final structure was validated by the program RAMPAGE [45].

The diffraction data are summarized in Tables 1 and 2. We confirmed that 345 water molecules and one tartaric acid molecule were in the asymmetric unit; they must have originated from the crystallization reagent. We could not recognize the clear electron densities as gadolinium ion, even at the final stage of the structure refinement. The details are available on the website of Protein Data Bank, the entry ID 8YK6.

## Discussion

### Applicability of magnetic levitation technique to protein crystallization

We used a paramagnetic solution of GdCl$_3$ to achieve magnetic levitation of HEWL crystals and thaumatin crystals. In the levitation of the HEWL crystals, the concentration of Gd$^{3+}$ ion was 0.388 mol/kg and the solution density $\rho_s$ was $1.09 \times 10^3$ kg/m$^3$ [35]. The magnetic flux density of

**Table 2. X-ray diffraction data of the thaumatin crystal.**

| Structure refinement | |
|---|---|
| Resolution range (Å) | 37.99–0.99 (1.01–0.99) |
| Completeness (%) | 99.9 (100.0) |
| σ Cutoff | None |
| No. of reflections, working set | 138,147 (10,046) |
| No. of reflections, test set | 7,248 (518) |
| Final $R_{cryst}$ | 0.123 (0.199) |
| Final $R_{free}$ | 0.133 (0.198) |
| No. of non-H atoms | |
| Total | 2226 |
| Protein | 1871 |
| Ligand | 10 |
| Solvent | 345 |
| R.M.S. deviations | |
| Bond lengths (Å) | 0.019 |
| Angles (°) | 2.312 |
| Average $B$ factors (Å$^2$) | |
| Protein | 11.65 |
| Ligand | 9.09 |
| Solvent | 24.18 |
| Ramachandran plot | |
| Most favored (%) | 98.0 |
| Allowed (%) | 2.0 |
| Outliers (%) | 0.0 |

$b_z$ was 2.2~2.5 T (5 T, $\phi$50 mm, Sumitomo, Co., Ltd.) [46] or 3.8 T (10 T, $\phi$100 mm, Japan Super-conductor Technology Inc. (JASTEC)) [28]. That is, the concentration of $Gd^{3+}$ ion was estimated as $0.388 \times 1.09 = 0.423$ M. In the case of the thaumatin crystals, as mentioned before in the crystallization process, the initial concentration of the paramagnetic substance of 0.40 M $Gd^{3+}$ ion was diluted to $70.0 / (70.0 + 283.5 + 175.0) = 0.132$ times, resulting in $0.40 \times 0.132 = 0.053$ M. This means that, if the other parameters of the thaumatin crystals were the same as those of the HEWL crystals, the magnetic force would be obliged to enhance to about $0.423 / 0.053 \approx 8.0$ times larger in anticipation of the levitation of the thaumatin crystals. Such a magnetic force can be realized when the magnitude of the magnetic flux density is increased by approximately $\sqrt{8.0} = 2.8$ times. Roughly speaking, a magnetic flux density of $2.8 \times 3.8 \approx 11$ T is required. A magnetic flux density of 13–14 T is necessary to hold the crystals stably. Such a theoretical estimation by using the HEWL crystallization condition was quantitatively close to the actual levitation of thaumatin crystals in 14.1 T. We surmise that a similar quantification may be applicable to the crystallization of other proteins by computing the approximate relation between the concentration of the paramagnetic substance (i.e., $Gd^{3+}$ ion) and the magnitude of the magnetic force. We would like to stress that magneto-Archimedes levitation could become a more practical technique to realize a containerless condition in protein crystallization.

## New findings by the structural analysis

Gadolinium was not confirmed in the thaumatin crystals in the structural analysis. In contrast, we could verify the presence of tartaric acid composed of carbon and oxygen, and 335 water molecules were identified very clearly. The disappearance of the image of gadolinium in the

electron density does not prove the complete absence of the gadolinium atoms in the crystal, but such a large difference between the gadolinium atom (atomic number 64) and the water molecule (molecular weight 18) is distinguishable in the images of electron density, and it is sufficient to make sure whether the gadolinium was bound to the thaumatin molecule. Our results support the finding that no gadolinium atoms were bound to the water molecules in the crystal.

There is a specific part of protein crystals called the "solvent region", the feature of which is greatly different from conventional crystals with low molecular compound. For the thaumatin crystal, 54% of the crystal volume is estimated as the solvent region, and other molecules, as well as water molecules, can easily come in and out of the region. By considering the isoelectric point, the thaumatin molecule has been found to have a positive charge [47,48]. Hence, we think it electrically difficult for thaumatin molecules and $Gd^{3+}$ ions to be attracted to each other. If by any chance gadolinium ions can get close to thaumatin molecules, such a state would be very rare and would be too difficult to detect. Thinking through the process of elimination, $Gd^{3+}$ ions enclosed in thaumatin crystals have little effect when conducting magneto-Archimedes levitation. However, the details of that matter are not clear, and we would like to leave that question to future research.

## Conclusions

We examined the magnetic effects of thaumatin crystals. We succeeded in realizing the magnetic levitation of thaumatin crystals by the technique of magneto-Archimedes effect, and demonstrated the crystal growth in situ under a containerless condition. These results are noteworthy in proposing that the containerless crystallization, which has only been successful with HEWL so far, is also applicable for thaumatin.

We observed the growing process of thaumatin chronologically and recorded the process in a short movie. We also presented a visualization of the magnetic orientation of thaumatin crystals under 6.0 T. In addition to that, we carried out a structural analysis of the thaumatin crystal. Through the refinement of the structure, we could not recognize the clear electron densities as gadolinium ions, supporting the fact that the magneto-Archimedes levitation of thaumatin is free from coordinate bonds to the paramagnetic substance of gadolinium. In other words, the paramagnetic substance of the aqueous precipitant solution may help only to promote the crystals' levitation, having little effect on thaumatin crystallization. Whatever happens, the magnetic levitation and containerless crystallization are highly versatile technologies and have potential for producing quality protein crystals. The findings on magnetic effects hereinabove also have academic value in contribuitng to the progress of new technical fields of protein crystal growth with pioneering engineering.

## Supporting information

**S1 Fig. The title of which is the "Short movie of Fig 4".** S1 Fig verified the magnetic levitation of thaumatin crystals by changing the magnitude of the magnetic force, where $b_z \frac{\partial b_z}{\partial z}$ was 1100 $T^2$/m in the vicinity of the upper coil edge. This movie was taken at 20˚C, and was recorded in real time speed. S1 Fig will be opened by the journal system of "Uploaded as supplementary information". Furthermore, S1 Fig is also opened in the website of the Library of Okayama University of Science (https://ous.repo.nii.ac.jp/records/2000296), as the repository. (WMV)

**S2 Fig. The title of which is the "Short movie of Fig 5".** S2 Fig made clear the chronological change of the crystal growth observed from the vessel top in the horizontal bore system, where

the magnetic flux (6.0 T) is directed upward in the picture. This movie was taken at 20˚C, and was edited at 20,000 times speed. S2 Fig will be opened by the journal system of "Uploaded as supplementary information". Furthermore, S2 Fig is also opened in the website of the Library of Okayama University of Science (https://ous.repo.nii.ac.jp/records/2000296), as the repository.
(MP4)

**S1 File.**
(DOCX)

## Acknowledgments

X-ray diffraction data were obtained using the beamlines BL41XU and BL45XU at SPring-8 (Hyogo, Japan) with the approval of JASRI, Japan Synchrotron Radiation Research Institute (Proposal Nos. 2021B2537, 2022A2537, 2022B2544, 2023A2544). We were supported by the Institute of Frontier Science and Technology, Okayama University of Science. We received gracious cooperation from the members of the Design and Manufacturing Center of Okayama University of Science for the development of the experimental equipment. Many experiments in this study, including preparatory research, were carried out with the support of the Center for Advanced High Magnetic Field Science in Osaka University under the Visiting Researcher's Program of the Institute for Solid State Physics, the University of Tokyo. We thank the National Science Center for Basic Research and Development, Hiroshima University for supplying liquid helium. This paper was proofread in English by Prof. Christopher Carman of the University of Occupational and Environmental Health. All the mentioned support was essentially important to the progress of our research project. We would like to express our deepest gratitude.

## Author Contributions

**Conceptualization:** Syou Maki.

**Data curation:** Syou Maki, Masao Fujiwara, Makoto Nakabayashi.

**Formal analysis:** Syou Maki, Makoto Nakabayashi, Kohji Inaka.

**Funding acquisition:** Syou Maki.

**Investigation:** Syou Maki, Makoto Nakabayashi, Seiichi Tanaka, Seiji Fujiwara, Kohji Inaka.

**Methodology:** Syou Maki, Masao Fujiwara, Yoshihisa Fujiwara, Kohji Inaka.

**Project administration:** Syou Maki, Kohji Inaka.

**Resources:** Syou Maki.

**Supervision:** Yoshihisa Fujiwara, Makoto Nakabayashi, Shotaro Morimoto, Seiichi Tanaka, Seiji Fujiwara, Masayuki Hagiwara, Kohji Inaka.

**Validation:** Syou Maki, Yoshihisa Fujiwara, Seiichi Tanaka, Masayuki Hagiwara.

**Visualization:** Syou Maki, Masao Fujiwara.

**Writing – original draft:** Syou Maki.

**Writing – review & editing:** Syou Maki, Masayuki Hagiwara.

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
