## [Decision Letter · Decision Letter 0]

9 Jun 2024

PONE-D-24-09811Magnetic effects of thaumatin crystalsPLOS ONE

Dear Dr. Maki,

Thank you for submitting your manuscript to PLOS ONE. After careful consideration, we feel that it has merit but does not fully meet PLOS ONE’s publication criteria as it currently stands. Therefore, we invite you to submit a revised version of the manuscript that addresses the points raised during the review process.

We look forward to receiving your revised manuscript.

Kind regards,

DongLu Shi, Ph.D

Academic Editor

PLOS ONE

“We developed all the observation systems and experimental procedures used in this study. The manufacturing and maintenance costs of the equipment were supported by MEXT/JSPS, KAKENHI, Grant Number JP15K04669 and Grant Number JP20K04335. X-ray diffraction data were obtained using the beamlines BL41XU and BL45XU at SPring-8 (Hyogo, Japan) with the approval of JASRI, Japan Synchrotron Radiation Research Institute (Proposal Nos. 2021B2537, 2022A2537, 2022B2544, 2023A2544). We also used research funding from the Institute of Frontier Science and Technology, Okayama University of Science. We received gracious cooperation from the members of the Design and Manufacturing Center of Okayama University of Science for the development of the experimental equipment. Many experiments in this study, including preparatory research, were carried out with the support of the Center for Advanced High Magnetic Field Science in Osaka University under the Visiting Researcher's Program of the Institute for Solid State Physics, the University of Tokyo. We thank the National Science Center for Basic Research and Development, Hiroshima University for supplying liquid helium. This paper was proofread in English by Prof. Christopher Carman of the University of Occupational and Environmental Health. All the mentioned support was essentially important to the progress of our research project. We would like to express our deepest gratitude.”

4. In the online submission form you indicate that your data is not available for proprietary reasons and have provided a contact point for accessing this data. Please note that your current contact point is a co-author on this manuscript. According to our Data Policy, the contact point must not be an author on the manuscript and must be an institutional contact, ideally not an individual. Please revise your data statement to a non-author institutional point of contact, such as a data access or ethics committee, and send this to us via return email. Please also include contact information for the third party organization, and please include the full citation of where the data can be found.

Reviewers' comments:

Reviewer's Responses to Questions

**Comments to the Author**

1. Is the manuscript technically sound, and do the data support the conclusions?

Reviewer #1: Yes

2. Has the statistical analysis been performed appropriately and rigorously? 

Reviewer #1: Yes

3. Have the authors made all data underlying the findings in their manuscript fully available?

Reviewer #1: Yes

4. Is the manuscript presented in an intelligible fashion and written in standard English?

Reviewer #1: Yes

5. Review Comments to the Author

Reviewer #1: The manuscript presents an innovative approach to protein crystallization using magneto-Archimedes levitation. The focus is on thaumatin crystals, with experiments conducted using a vertical bore system for levitation and a horizontal bore system for orientation. The study also includes structural analysis of the crystallized thaumatin. The manuscript demonstrates significant advances in the field and provides comprehensive theoretical and experimental details.

Strengths:

Innovative Approach: The use of magneto-Archimedes levitation for crystallization is novel and well-executed, showing potential for broader applications in structural biology.

Detailed Methodology: The manuscript provides thorough descriptions of the experimental setups, including the magnetic field parameters, crystallization conditions, and structural analysis techniques.

Clear Visual Evidence: The chronological photographic documentation of thaumatin crystallization provides clear evidence of the process and the effects of magnetic forces on crystal growth.

Revisions Needed:

Clarification of Motivation: The introduction should better clarify why thaumatin was chosen beyond its industrial sweetness. Emphasize its significance as a model protein in structural biology and its relevance to thermal property measurements.

Literature Review: Expand the literature review to include more studies on magnetic effects on protein crystallization, especially those related to thaumatin. This will better contextualize the novelty of your work.

Hypothesis and Objectives: Clearly state the hypothesis and main objectives at the end of the introduction. This will guide the reader on what to expect in the subsequent sections.

Methodology: Detailed Experimental Protocols: Provide more specific details about the preparation of the buffer solution and the concentration of the precipitant agents used for crystallization. Include information on how the magnetic field gradient was measured and controlled.

Reproducibility: Include a more detailed description of the experimental conditions to ensure reproducibility. This should cover the exact specifications of the superconducting magnet and any calibration procedures.

Data Presentation: Figures 3 and 4 should be accompanied by more quantitative data, such as measurements of crystal size and growth rates over time. Including graphs or charts would enhance the clarity of the results.

Statistical Analysis: If applicable, provide statistical analysis to support the findings, particularly when comparing the effects of different magnetic field strengths on crystal growth.

Comparison with Previous Work: Discuss the results in the context of previous studies on HEWL and other proteins. Highlight what new insights your study brings to the field.

Mechanistic Insights: Provide a more in-depth discussion of the underlying mechanisms of magneto-Archimedes levitation and magnetic orientation. Explain why the effects observed in thaumatin crystals differ from those in HEWL.

Conclusion:

The manuscript is well-written and presents a significant advancement in the field of protein crystallization. With the suggested revisions, it will provide a clearer, more comprehensive, and impactful contribution to the scientific community.

6. PLOS authors have the option to publish the peer review history of their article (what does this mean?). If published, this will include your full peer review and any attached files.

Reviewer #1: No

---

## [Author Response · Author response to Decision Letter 0]

20 Jul 2024

Response to Reviewers, PONE-D-24-09811

Title

Magnetic effects of thaumatin crystals; observation of crystal growth by magneto-Archimedes levitation and magnetic orientation

The revised parts in response to the comments by the reviewer are highlighted in red.

The revised parts by the authors are highlighted in blue.

The revised parts where the text is simply moved are highlighted in green.

The revised parts according to the Journal editors’ requirements are highlighted in pink.

Reviewer’s comments

[1] Clarification of Motivation: The introduction should better clarify why thaumatin was chosen beyond its industrial sweetness. Emphasize its significance as a model protein in structural biology and its relevance to thermal property measurements.

[Res]

The reasons why we chose thaumatin are that the crystals grow large enough to be visible to the naked eye, its structure is known, and there is little previous research on the effects of magnetic fields. Another major reason is that the transient short hot-wire method can be applied. These were already mentioned in the MS. Only HEWL has been measured for its thermal properties so far. If the thermal properties of another model protein crystal with a known structure are clarified, it will be possible to structurally compare the differences in thermal properties with HEWL and to deeply understand the heat transfer performance from a molecular structural perspective. This will innovate the temperature control system for crystal growth and greatly advance the industrial technology.

In the revised version, the following description has been added to the middle of the introduction.

Page 4, lines 12-15 from the top:

“If the thermal properties of thaumatin are comparable to those of HEWL, instructive information on heat transfer of other protein crystals can be distinguished by using a molecular structural perspective. This will make a contribution to the progress of industrial technology for controlling crystal growth.” 

[2] Literature Review: Expand the literature review to include more studies on magnetic effects on protein crystallization, especially those related to thaumatin. This will better contextualize the novelty of your work.

[Res]

We have already cited most of the research papers on the magnetic effects on thaumatin crystallization, but after a detailed search we found one more paper that is related to thaumatin crystallization under microgravity condition. We newly added that paper in the revised version.

13. Lorber B., Sauter C., Robert M.C., Capelle B., Giegé R., Crystallization within agarose gel in microgravity improves the quality of thaumatin crystals. Acta Crystallogr D Biol. Crystallogr. 1999, 55: 1491–1494. DOI: 10.1107/s0907444999008902

[3] Hypothesis and Objectives: Clearly state the hypothesis and main objectives at the end of the introduction. This will guide the reader on what to expect in the subsequent sections.

[Res]

This research is not in the fields of statistics, medicine, or epidemiology, so inductive argumentation, in which a hypothesis is presented and then verified, is not suitable. However, we agree with the point that some comments clarifying our research purpose should be included in the latter half of the introduction.

In the revised version, we added the following sentence about what can be expected if magnetic levitation is realized to thaumatin. This is a comprehensive presentation of the future vision of this research, as clarifying the research objectives.

Page 5, lines 8-10 from the top:

“If magnetic levitation can be realized in thaumatin crystal growth, we will establish that our method, which has only been successful with HEWL so far, is applicable to the crystallization of other proteins.” 

[4] Methodology: Detailed Experimental Protocols: Provide more specific details about the preparation of the buffer solution and the concentration of the precipitant agents used for crystallization. Include information on how the magnetic field gradient was measured and controlled.

[Res]

The procedure for preparing the solution for crystallization is to mix three types of solutions in the optimal concentration ratio. It may seem complicated, but it is actually very simple. The concentration conditions were mentioned in the section of Crystallization in page 11.

On the other hand, the concentration of the paramagnetic subject (gadolinium chloride), which works as a driving force on magnetic levitation, is very important. This is necessary to verify whether the levitation occurs or not. Therefore, the concentration after mixing was calculated in the Discussion.

In the revised version, the three types of solutions are labeled as (a), (b), and (c), and the following description has been added. We believe this revision will help readers understand.

Page 11, lines 3-4 from the top:

“All we had to do was to mix three types of solutions, (a) protein, (b) buffer, and (c) precipitant, in the optimal volume ratio.”

The magnetic flux density and magnetic field gradient were referenced in the performance table attached to the magnet. The following description has been added in the revised version.

Page 11, lines 10-12 from the top:

“The magnetic conditions of br, b�, bz and in the bore were referenced in the performance table attached to the magnet.” 

[5] Reproducibility: Include a more detailed description of the experimental conditions to ensure reproducibility. This should cover the exact specifications of the superconducting magnet and any calibration procedures.

[Res]

We agree to the reviewer's request, but it is difficult to ensure reproducibility with biological samples. Protein crystallization is not always reproducible. This is thought to be due to protein degradation or denaturation, but sometimes microcrystals or precipitation occurs. As a countermeasure, we used crystals precipitated in the absence of a magnetic field to confirm in advance whether the crystals would magnetically levitate while observing them with the CCD camera in Fig. 1(c). The magnetic levitation is highly reproducible regardless of the deterioration of protein, because it depends on the magnetic susceptibility and the strength of the magnetic field. Owing to the preliminary experiments in advance, we were able to achieve highly reproducible magnetic levitation on the crystal growth.

In the revised version, the following comments have been added after the final paragraph on page 11 of the manuscript.

“We used crystals precipitated in the absence of a magnetic field to confirm in advance whether the crystals would magnetically levitate, then we observed the magnetic levitation of those crystals with the CCD camera in Fig. 1(c). The levitation is highly reproducible regardless of the deterioration of protein, because it depends simply on the magnetic susceptibility and the magnetic field strength. Owing to such preliminary experiments, we were able to achieve highly reproducible levitating crystal growth.”

In addition, the following sentence has been added to the Appendix on page 21 of the MS.

“Levitated crystals are marked with “Lev”, and non-levitated crystals are marked with “N.L.”.”

[6] Data Presentation: Figures 3 and 4 should be accompanied by more quantitative data, such as measurements of crystal size and growth rates over time. Including graphs or charts would enhance the clarity of the results.

[Res]

Thank you for the advice. We added some sentences presenting the crystal size and growth rates. The revised parts are underlined.

From the 12th line from the top of page 12 to the 6th line from the top of page 13, the section of “Magnetic levitation of thaumatin crystals”

“Figure 3(b) shows the growing thaumatin crystals at the meniscus one day later. They were approximately 0.1-0.2 mm in size by appearance. This image presents clear evidence that it is possible to levitate thaumatin crystals by the magneto-Archimedes effect, similar to the mechanism in HEWL crystals. Figure 3(c) shows the state after four days, revealing that many crystals appeared at the interface as growing. They were approximately 0.3-0.5 mm in size. Figures 3(d), 3(e), and 3(f) are the images after 7, 21 and 48 days, respectively. It was difficult to measure the size of each crystal. The thickness of the crystals further increased to be as thick as to cover up the whole meniscus. Based on these images, the thickness in Figs. 3(d) and 3(e) were about 0.8 and 0.9 mm, respectively, and the maximum thickness in Fig. 3(f) exceeded 1 mm after 48 days. The crystal growth rate was a maximum of 0.2 mm/d up to the first day, and a maximum of 0.1 mm/d from the first to fourth days, but then gradually slowed down, reaching a maximum of about 0.01 mm/d from the 21st to 48th days.”

[7] Statistical Analysis: If applicable, provide statistical analysis to support the findings, particularly when comparing the effects of different magnetic field strengths on crystal growth.

[Res]

Statistical analysis is not applicable in this study.

[8] Comparison with Previous Work: Discuss the results in the context of previous studies on HEWL and other proteins. Highlight what new insights your study brings to the field.

[Res]

As for the reviewer’s comment of “Comparison with Previous Work”, we cannot conduct a comparison of the results because there is almost no analogous research on this research. Regarding the magnetic levitation, the only one we have succeeded in is magnetic levitation growth of lysozyme crystals, which we achieved 20 years ago.

As for the reviewer’s comment of “Highlight what new insights your study brings”, we would like to respond by slightly amending the conclusion. In the revised version, Conclusion was partially modified as: 

Page 19-20.

“We examined the magnetic effects of thaumatin crystals. We succeeded in realizing the magnetic levitation of thaumatin crystals by the technique of magneto-Archimedes effect, and demonstrated the crystal growth in situ under a containerless condition. These results are noteworthy in proposing that the containerless crystallization, which has only been successful with HEWL so far, is also applicable for thaumatin. 

We observed the growing process of thaumatin chronologically and recorded the process in a short movie. We also presented a visualization of the magnetic orientation of thaumatin crystals under 6.0 T. In addition to that, we carried out a structural analysis of the thaumatin crystal. Through the refinement of the structure, we could not recognize the clear electron densities as gadolinium ions, supporting the fact that the magneto-Archimedes levitation of thaumatin is free from coordinate bonds to the paramagnetic substance of gadolinium. In other words, the paramagnetic substance of the aqueous precipitant solution may help only to promote the crystals’ levitation, having little effect on thaumatin crystallization. Whatever happens, the magnetic levitation and containerless crystallization are highly versatile technologies and have potential for producing quality protein crystals. The findings on magnetic effects hereinabove also have academic value in contribuitng to the progress of new technical fields of protein crystal growth with pioneering engineering.”

[9] Mechanistic Insights: Provide a more in-depth discussion of the underlying mechanisms of magneto-Archimedes levitation and magnetic orientation. Explain why the effects observed in thaumatin crystals differ from those in HEWL.

[Res]

That is also explained in the section on Magneto-Archimedes levitation in Theoretical Ideas. More details on the magneto-Archimedes levitation have also been described in our previous study [21]. The levitation principle observed in thaumatin crystals was almost the same as that in HEWL. The precipitation styles were also very similar to each other. The only difference was the magnitude of paramagnetic susceptibility in the protein solution. As proof of this, the meniscus of the thaumatin solution in Fig. 3 was not horizontal. This is because the paramagnetic susceptibility of thaumatin solutions was low. In the case of HEWL, the solution is strongly attracted downward by the magnetic force, so the meniscus is deformed and becomes almost horizontal in the magnetic field. 

The revised version adds the following statement in the section of Magnetic levitation of thaumatin crystals.

Page 12, lines 9-12 from the top:

“In the case of lysozyme levitation, the concentration of paramagnetic substance in the solution was high, so the solution was strongly pulled downward by the magnetic force, causing the meniscus to deform and became almost horizontal in the magnetic field [22].”

In addition, the revised version adds the following description about magnetic orientation in the section of Magnetic orientation of thaumatin crystals.

Page 15, lines 9-11 from the top:

“Since all the crystals were grown oriented from the beginning, the magnetic orientation of thaumatin is found to be possible at 6 T, smaller than 10-11 T.”

Page 15, lines 12-13 from the top:

“We assume that the levitated crystals must be oriented, although the presence of the orientation can not be observed in the photograph of Fig. 3.”

Journal editors’ requirements

[10] Funding Information and Financial Disclosure sections

We note that the grant information you provided in the ‘Funding Information’ and ‘Financial Disclosure’ sections do not match. When you resubmit, please ensure that you provide the correct grant numbers for the awards you received for your study in the ‘Funding Information’ section.

[Res]

Sorry for the confusion. We did not receive any fundings from the US government or international research institutions for this research, but we did receive academic support from the Japanese government (MEXT/JSPS, KAKENHI). We removed the information about the research grant from the attachment, and listed it in the Financial Disclosure in Funding Information.

MEXT/JSPS, KAKENHI, Grant Number JP15K04669.

MEXT/JSPS, KAKENHI, Grant Number JP20K04335.

[11] In the online submission form you indicate that your data is not available for proprietary reasons and have provided a contact point for accessing this data. Please note that your current contact point is a co-author on this manuscript. According to our Data Policy, the contact point must not be an author on the manuscript and must be an institutional contact, ideally not an individual. Please revise your data statement to a non-author institutional point of contact, such as a data access or ethics committee, and send this to us via return email. Please also include contact information for the third party organization, and please include the full citation of where the data can be found.

[Res]

The comment, “All the data in this study are available from the corresponding author upon reasonable request” is often seen in the journals. Now the contact point was changed to the research institute at the university to which the corresponding author (Syou Maki) belongs.

[12] Please include captions for your Supporting Information files at the end of your manuscript, and update any in-text citations to match accordingly. 

[Res]

We changed it as the editors said.

Page 22:

“We will make two short movies available to the public. “Short movie of Figure 4” verified the magnetic levitation of thaumatin crystals by changing the magnitude of the magnetic force, where was 1100 T2/m in the vicinity of the upper coil edge. This is recorded in real time speed.

The other, “Short movie of Figure 5”, made clear the chronological change of the crystal growth observed from the vessel top in the horizontal bore system, where the magnetic flux (6.0 T) is directed upward in the picture. This movie is edited at 20,000 times speed. Both movies were taken at 20�C. ”

In addition, we added a comment about these short movies in the caption of Figure 4 and Figure 5.

Others

[13] Figure 1(b) was modified.

[14] In the Introduction, we added the following sentence to present why we attempted the structural analysis of thaumatin crystals.

Page 5, lines 12-16 from the top:

“As far as we know, there has been no previous research on using a paramagnetic substance to crystallize thaumatin, and it is also unknown wheth

---

## [Decision Letter · Decision Letter 1]

13 Aug 2024

PONE-D-24-09811R1Magnetic effects of thaumatin crystalsPLOS ONE

Dear Dr. Maki,

Thank you for submitting your manuscript to PLOS ONE. After careful consideration, we feel that it has merit but does not fully meet PLOS ONE’s publication criteria as it currently stands. Therefore, we invite you to submit a revised version of the manuscript that addresses the points raised during the review process.

We look forward to receiving your revised manuscript.

Kind regards,

Amir Elzwawy, Ph.D.

Academic Editor

PLOS ONE

Journal Requirements:

Reviewers' comments:

Reviewer's Responses to Questions

**Comments to the Author**

1. If the authors have adequately addressed your comments raised in a previous round of review and you feel that this manuscript is now acceptable for publication, you may indicate that here to bypass the “Comments to the Author” section, enter your conflict of interest statement in the “Confidential to Editor” section, and submit your "Accept" recommendation.

Reviewer #1: All comments have been addressed

Reviewer #2: All comments have been addressed

Reviewer #3: (No Response)

2. Is the manuscript technically sound, and do the data support the conclusions?

Reviewer #1: Yes

Reviewer #2: Yes

Reviewer #3: Yes

3. Has the statistical analysis been performed appropriately and rigorously? 

Reviewer #1: Yes

Reviewer #2: Yes

Reviewer #3: N/A

4. Have the authors made all data underlying the findings in their manuscript fully available?

Reviewer #1: Yes

Reviewer #2: Yes

Reviewer #3: Yes

5. Is the manuscript presented in an intelligible fashion and written in standard English?

Reviewer #1: Yes

Reviewer #2: Yes

Reviewer #3: No

6. Review Comments to the Author

Reviewer #1: The authors have addressed all the issues and I recommend it to be accepted for publication in PLOS ONE.

Reviewer #2: The study has shown spectacular results. Congratulations, since the improvements have been made, it is recommended to accept it.

Reviewer #3: Comment for authors:

This study represents the first demonstration of thaumatin crystallization using Archimedes magnetic levitation. It investigates how magnetic fields affect the growth and orientation of these protein crystals. To achieve this aim, the authors used gadolinium chloride as a paramagnetic agent in a strong magnetic field generated from a superconducting magnet. This approach can be used to produce high-quality protein crystals and can also pave the way for exploring the thermal properties of thaumatin. However, this paper needs some improvement in terms of language, as well as some minor adjustments.

1. To enhance readability the authors should simplify the complex sentences, and eliminate unnecessary terminology, such as "strictly speaking,".

2. Please reorganize the introduction into paragraphs that flow logically from the background to specific research questions.

3. Please provide the full expansion of the abbreviation PDB in the introduction.

4. The introduction would benefit from a more comprehensive review of the relationship between understanding the thermal properties of proteins and the development of crystal growth technology

5. In line 59, the authors should cite studies that confirm the presence of magnetic orientation of thaumatin.

6. The role of paramagnetic substances is not clear in the introduction.

7. To ensure consistency, I suggest that authors follow a uniform approach when referencing figures in the text.

8. Please explain in the paper how you calculated the magnetic field and the magnetic force in FIG1(d), and 1(e).

9. Please provide appropriate citations to support the statement in line 153 "In most previous studies, magnetic orientation of thaumatin crystals was examined in a small capillary tube".

7. PLOS authors have the option to publish the peer review history of their article (what does this mean?). If published, this will include your full peer review and any attached files.

Reviewer #1: No

Reviewer #2: **Yes: **Mohd Ridha Muhamad

Reviewer #3: No

---

## [Author Response · Author response to Decision Letter 1]

22 Nov 2024

Response to Reviewers, PONE-D-24-09811

Title

Magnetic effects of thaumatin crystals; observation of crystal growth by magneto-Archimedes levitation and magnetic orientation

The revised parts in response to the comments by the Reviewer 3 are highlighted in red.

The revised parts by the authors are highlighted in blue.

Revisions by the comments of the Reviewer #3: Comment for authors:

[1] To enhance readability the authors should simplify the complex sentences, and eliminate unnecessary terminology, such as “strictly speaking”.

[Res.]

In the previous version, we used the phrase “strictly speaking” only once, in the line 103 of page 7 for the explanation of the roles of the component fr. As we mentioned in the text, few magnetic levitation studies have focused on the effect of fr. This is because the magnitude of fz is overwhelmingly larger than that of fr in a conventional solenoidal superconducting magnet. However, we think the effect of fr is very important practically. We observed that even a small force works as a definite driving force in a pseudo-weightless state. In my previous research on magnetic separation of sucrose and glucose, we could visualize an interesting phenomenon in which those particles fall along an hourglass-like trajectory due to the influence of fr in the weightless state during the fall [30]. We also achieved the first successful magnetic levitation of crystals by tilting the bore horizontally and using a vertical fr [31]. Thus, the role of fr cannot be ignored, and therefore we think there is no need to delete the phrase “strictly speaking” in the text. However, it might be possible that the reviewer felt that phrase too strong, so we changed it to “More particularly”, instead. 

In addition, we also added the following sentence in Page 7, Lines 113-116, and newly used some papers of Ref [30, 31] to the reference.

“We confirmed that the effect of fr could become tangible through the techniques of magnetic separation in the weightless condition [30] as well as the experiment for thermal properties of HEWL crystals [31]”

30. Maki S., Hirota N., Magnetic separation technique on binary mixtures of sorbitol and sucrose, J. Food Eng. 2014, 120 C: 31–36, DOI: 10.1016/j.jfoodeng.2013.07.006

31. Maki S., Tanaka S., Miyagi K., Mori T., Isaka Y., Hagiwara M., Fujiwara S., Thermal conductivity and thermal diffusivity of lysozyme crystals, the c-axis of which is magnetically orientated along the direction of the probe wire, Exp. Heat Transf. 2023, 36 (4), DOI: 10.1080/08916152.2023.2197903

[2] Please reorganize the introduction into paragraphs that flow logically from the background to specific research questions.

[Res.]

One of the main results of this paper is the successful magnetic levitation of thaumatin. However, the introduction did not explain much about the significance of magnetic levitation of protein crystals. We realized that we should explain more about the significance and engineering advantages of magnetic levitation of protein crystal growth. This would make the intent of this paper more easily understandable to readers. Therefore, we added the following sentence.

Pages 4-5, Lines 58-64.

“Growing protein crystals by levitation has an advantage other than the measurement of thermal properties. Generally, protein crystals are heavier than the solution, the crystals grow at the bottom of the container and often adhere to the bottom wall. The gas-liquid interface is a containerless condition [12-16], and it is expected that high-quality crystals will be produced. By growing them at the interface, we can easily pick them up without damaging them, which is convenient for structural analysis.”

[3] Please provide the full expansion of the abbreviation PDB in the introduction.

[Res.]

Thank you for the advice. We revised the sentence as follows:

Page 4, Lines 46-47.

“Crystallographic details of thaumatin are made public in the Protein Data Bank (see the data of PDB entry 1RQW)”.

[4] The introduction would benefit from a more comprehensive review of the relationship between understanding the thermal properties of proteins and the development of crystal growth technology.

[Res.]

Thank you for the advice. Proteins are well known to have a very high degree of supersaturation relative to their saturated solubility curves. Especially in that supersaturated state, temperature control has a significant impact on the crystal growth. Hence, we can deduce that thermal properties of protein crystal are essentially important knowledge for the progress of crystal growth technology.

In the revised version, we will cite a research paper by a group at Osaka University that developed an accurate temperature control machine, named TAON. This paper demonstrated that protein crystallization is clearly changed by the temperature. (The year after this paper was published, Maki transferred to this laboratory and made some contributions to the development of this research.)

Pages 4, Lines 55-58, the following sentence has been added:

“Such instruction will be useful for the development of crystal growth technology because we notice that temperature control has a significant impact on the crystallization [11] and thermal properties can make a contribution to the progress of temperature control.”

11. Murai R., Nakata S., Kashii M., Adachi H., Niino A., Takano K., Matsumura H., Murakami S., Inoue T., Mori Y., Sasaki T., Cooling-rate screening system for determining protein crystal growth conditions: J. Cryst. Grow. 2006, 292: 434–436, DOI: 10.1016/j.jcrysgro.2006.04.050

[5] In line 59, the authors should cite studies that confirm the presence of magnetic orientation of thaumatin.

[Res.]

In the previous version, we cited some research just before the sentence in page 5. Then, we omitted to present them there. In the new version, we refer to the papers of [19, 20] in Page 5, Line 69.

[6] The role of paramagnetic substances is not clear in the introduction.

[Res.]

The magneto-Archimedes effect is realized where the environment surrounding a diamagnetic substance is made paramagnetic and the repulsion of the magnetic forces acting on each other is used as magnetic levitation of diamagnetic substance (i.e., protein crystals). The amount of paramagnetic substance strengthens the repulsion, and the greater the amount (higher concentration), the smaller the magnetic force that can levitate the diamagnetic object. In actuality, protein crystals tend to aggregate with a large amount of paramagnetic substance. Thaumatin crystallizes when a smaller amount of paramagnetic substance (GdCl3) is added than HEWL. Therefore, the only option was to strengthen the magnetic force.

The mechanism of the levitation is explained in detail in the section of “Magneto-Archimedes levitation”; hence we cannot devote many pages to explaining it in the introduction. 

In Page 5, Line 75, we added this short sentence: 

“with a small amount of paramagnetic substance”

[7] To ensure consistency, I suggest that authors follow a uniform approach when referencing figures in the text.

[Res.]

Thank you for the comment. In the previous version, we were consistent in using “Figure” at the beginning of a sentence and “Fig. or Figs.” in the middle of the sentence. In the present version, we will standardize everything to “Figure”. 

[8] Please explain in the paper how you calculated the magnetic field and the magnetic force in FIG1(d), and 1(e).

[Res.]

The distributions of B and F in Figure 1 and in Figure 2 were calculated with Maki’s original program code of FORTRAN, developed in 2003. This code can simulate the magnetic field distributions around solenoidal circular coils. That is, this program approximates the actual shape of a superconducting magnet to that of a solenoidal electromagnet. Maki has utilized this program for many years and sometimes refined the code to progress his research on the magneto science.

To begin with, the magnet bore is divided into an equally spaced grid in a cylindrical coordinate system, and the center of each grid is defined as a scalar point. Next, a single circular coil is assumed to simulate a superconducting magnet coil. The one-turn circular coil is divided into equal infinitesimal line elements. The magnetic flux density acting between the line element and the scalar point is calculated using the Biot-Savart law. By calculating in the radial, circumferential, and axial directions, the magnetic flux density vector can be calculated. This process is repeated between each infinitesimal line element and its scalar point. This calculates the magnetic flux density vector B created by the one-turn circular coil at that scalar point. Since the calculation is expressed as a complex triple integral formula, the Simpson method is used for analytical approximation.

By assuming multiple circular coils with different diameters and positions, the magnetic field with multi-layer coils can be simulated. Its calculation process is similar to the case of a one-turn circular coil. By changing the scalar point, it is possible to calculate the magnetic field vectors even if the vessel is set up to an off-centered position from the central axis of the bore. Finaly, the magnetic force vector F is calculated using the magnetic flux density vector B.

The actual shape and size of a commercial superconducting magnet are concealed by industrial secrets, and its detailed structure has not been made public. The calculation results by our program code are almost identical to the actual performance. The first research with this code is shown below. We will add this paper to the references. The following explanation has been added to the main text.

Pages 9-10, Lines 154-157.

“The distribution of the magnetic flux density vector B and the magnetic force vector F were obtained by the calculations using an original calculation code developed by the authors. In this program, the magnetic flux density is calculated using the Biot-Savart law [36]. Details are explained in previous studies [37, 38].”

36. Landau L. D., Lifshitz E. M., The Classical Theory of Fields 4th Edition, Course of Theoretical Physics Vol. 2, Pergamon Press, p. 103, ISBN 0-08-025072-6.

37. Maki S., Ataka M., Suppression and promotion of convection in water by use of radial components of the magnetization force, J. Appl. Phys. 2004, 96: 1696–1703, DOI: 10.1063/1.1763239

38. Maki S., Ataka M., Effects of non-axisymmetric magnetization force on natural convection of water at various off-centered positions in a superconducting magnet: Numerical computation studies, Jpn. J. Appl. Phys. 2005, 44: 1132–1138, DOI: 10.1143/jjap.44.1132

[9] Please provide appropriate citations to support the statement in line 153 “In most previous studies, magnetic orientation of thaumatin crystals was examined in a small capillary tube”.

[Res.]

Thank you for the advice. We provided appropriate citations in the first sentence of the section “Equipment for the magnetic orientation” in Page 10, Line 170.

Other revisions

[10] Six new papers have been cited. The citation numbers have been revised accordingly.

[11] The captions for Figures 1 and 2 have been updated to explain the derivation of the numerical results.

---

## [Editor Report · Decision Letter 2]

25 Nov 2024

Magnetic effects of thaumatin crystals

PONE-D-24-09811R2

Dear Dr. Maki,

We’re pleased to inform you that your manuscript has been judged scientifically suitable for publication and will be formally accepted for publication once it meets all outstanding technical requirements.

Kind regards,

Amir Elzwawy, Ph.D.

Academic Editor

PLOS ONE

Additional Editor Comments (optional):

Dear authors, the journey of your publication is about to reach the desired destination. The peer reviewing process and editorial stages took a lot of time for considering multiple points and for finding suitable reviewers. Now, I feel the manuscript is much improved and can be accepted. Congratulations
---

## [Editor Report · Acceptance letter]

28 Nov 2024

PONE-D-24-09811R2 

PLOS ONE

Dear Dr. Maki, 

I'm pleased to inform you that your manuscript has been deemed suitable for publication in PLOS ONE. Congratulations! Your manuscript is now being handed over to our production team.

Kind regards, 

on behalf of

Dr Amir Elzwawy 

Academic Editor

PLOS ONE